A novel fast pedestrian recognition algorithm based on point cloud compression and boundary extraction

Zhang Yanjun yj.zhang@zspt.edu.cn
Zhong Shan Polytechnic , Zhongshan , China
Ahmed Imran
Electronic publication date: 2023 Jun 16
Publication date: 2023
Volume: 9
Electronic Location ID: e1426
Received 2023 Mar 30; Accepted 2023 May 14
Copyright: © 2023 Zhang
Copyright year: 2023
Copyright holder: Zhang
License: This is an open access article distributed under the terms of the Creative Commons Attribution License, which permits unrestricted use, distribution, reproduction and adaptation in any medium and for any purpose provided that it is properly attributed. For attribution, the original author(s), title, publication source (PeerJ Computer Science) and either DOI or URL of the article must be cited.
License URL: https://creativecommons.org/licenses/by/4.0/

Keywords: Driverless, Boundary extraction, Point cloud data compression, Pedestrian identification

Funding: Research on Key Technologies of Encoding and Decoding of Real-time Vehicular Lidar Point Cloud Sequences 2021ZDZX1123 This study was supported by the Guangdong Provincial Department of Education New Generation Information Technology Key Special Field Fund project: Research on Key Technologies of Encoding and Decoding of Real-time Vehicular Lidar Point Cloud Sequences (No. 2021ZDZX1123). The funders had no role in study design, data collection and analysis, decision to publish, or preparation of the manuscript.

==============================
Reason

Pedestrian recognition has great practical value and is a vital step toward applying path planning and intelligent obstacle avoidance in autonomous driving. In recent years, laser radar has played an essential role in pedestrian detection and recognition in unmanned driving. More accurate high spatial dimension and high-resolution data could be obtained by building a three-dimensional point cloud. However, the point cloud data collected by laser radar is often massive and contains a lot of redundancy, which is not conducive to transmission and storage. So, the processing speed grows slow when the original point cloud data is used for recognition. On the other hand, the compression processing of many laser radar point clouds could save computing power and speed up the recognition processing.

Methodology

The article utilizes the fusion point cloud data from laser radar to investigate the fast pedestrian recognition algorithm. The focus is to compress the collected point cloud data based on the boundary and feature value extraction and then use the point cloud pedestrian recognition algorithm based on image mapping to detect pedestrians. This article proposes a point cloud data compression method based on feature point extraction and reduced voxel grid.

Results

The Karlsruhe Institute of Technology and Toyota Technological Institute data set is used to investigate the proposed algorithm experimentally. The outcomes indicate that the peak signal-to-noise ratio of the compression algorithm is improved by 6.02%. The recognition accuracy is improved by 16.93%, 17.2%, and 16.12%, corresponding to simple, medium, and difficult scenes, respectively, when compared with the point cloud pedestrian recognition method based on image mapping, which uses the random sampling method to compress the point cloud data.

Conclusion

The proposed method could achieve data compression better and ensure that many feature points are retained in the compressed Point Cloud Data (PCD). Thus, the compressed PCD achieves pedestrian recognition through an image-based mapping recognition algorithm.

Introduction

Pedestrian recognition technology uses computer vision technology to determine whether pedestrians exist in an image or a video sequence and then accurately locate them. So, it detects pedestrians’ locations and measures their distances. Thus, autonomous vehicles could be run safely. Pedestrian recognition technology can effectively protect pedestrians’ safety and reduce the number of traffic accidents and injuries by avoiding collisions since an abrupt or continuous flow of pedestrians could exist on roads at any time. Therefore, it helps autonomous driving make sound decisions. In recent years, with the development of vehicle-mounted recognition equipment, pedestrian recognition technology has also been further advanced. Special functional cameras, such as binocular cameras, obtain images that can sense information in three dimensions and realize both image processing and recognition. Also, three-dimensional spatial information could be obtained more accurately through laser scanning. Moreover, compared with the conventional two-dimensional image recognition method, PCD collected by 3D laser radar overcomes the limitations of 2D and becomes closer to the original representation of objects. However, due to the different degrees of design defects of the lidar acquisition equipment, external factors such as light, air vibration, and dust, as well as the influence of how competent the operator would be, the collected PCD has several problems such as more or fewer outliers, lack of sub-data within the region, data distortion and redundancy (Hu & Zhao, 2013).

The hardware system of 3D laser radar technology has become more advanced. However, the PCD collected by laser radar still has the characteristics of high redundancy, nonlinear error distribution, and incompleteness, which brings great difficulties to the intelligent processing of massive 3D PCD. Thus, the post-processing research of PCD still has great deficiencies (Yiming et al., 2014; Bisheng, Fushun & Ronggang, 2017; Xinlian et al., 2005), such as its intelligent processing level being relatively backward, its software interface being unfriendly, the data interface of professional applications, and other aspects requiring improvements. In addition, the equipment needs to collect as much information as possible on the physical surface when data collection occurs. Even though the accuracy of the point cloud collection equipment continues to improve, the final data volume often grows very large, the density is very high, and the problem of data redundancy is relatively severe, whose sizes are often at the Gigabyte, even reaching the Terabyte levels, which makes the operation of PCD storage and transmission lead to high requirements on computer hardware (Yiling et al., 2018; Zhang, Cheng & Liu, 2011; Jingzhong, Youchuan & Shengwang, 2008; Dehai, Hao & Wei, 2012). So, PCD must be compressed correctly in practical applications while maintaining the model’s accuracy as much as possible. More up-to-date research can be found in Garrote et al. (2022), Cura, Perret & Paparoditis (2017) and Li et al. (2015).

State-of-the-art studies

The compression of PCD has attracted the attention of many scholars, and several experimental research has been carried out that have resulted in various proposed compression methods. The problem of an excessive amount of PCD has been dramatically resolved. Moreover, PCD could reflect the structure and shape of physical entities because spatial information and three-dimensional structure information of objects in the surrounding environment are contained. Thus, feature extraction methods can be employed to expand the high-dimensional feature information of PCD, which fits 3D object recognition in scenes. Therefore, studies to utilize PCD better for recognition purposes have also become a research hot spot in recent years.

In the research to compress PCD, Zhang, Cheng & Liu (2011) improved a compression method for feature retention, mainly based on the bounding box and octree theory. The point cloud features were saved by calculating the average vector and curvature information. This method could achieve data compression but did not intentionally preserve the boundary feature points and retained only the limited feature points in the non-boundary region.

Morell et al. (2014) proposed a lossy compression system based on plane extraction. This method represented the plane points of each scene with the Delaunay triangulation and a set of point/region information, which could achieve different compression rates or accuracy rates for the dataset. However, this method lacked attention to feature points, resulting in the data points retained after compression not necessarily being feature points. Chenghui & Xiping (2016) proposed a skewed height deviation compression method based on the massive hierarchical PCD. The point cloud was divided into several levels, converted into a linear point cloud, and then compressed by the skewed height deviation threshold. This method had advantages in retaining richer feature points under a higher compression rate. The quality of PCD compression has been related to the hierarchical situation of the point cloud. However, the calculation became more complex, and the delay was more prolonged.

Park & Jang (2018) proposed a method that could quickly calculate the point cloud distortion to measure the compression sequence’s error. Therefore, an improved search method was presented. Although this algorithm had certain speed advantages, it retained less detailed feature points. Renzhong et al. (2017) established a three-dimensional voxel grid for the point cloud and simplified each three-dimensional voxel grid according to the estimated average vector. The method could achieve uniform point cloud compression but could not retain more feature points in areas with significant feature changes. Chenhang et al. (2020) proposed a point cloud compression algorithm based on vector similarity and a point cloud restoration algorithm, CVS, to resolve the problems related to retaining the difficult detailed features in point cloud compression and the holes in the flat area of the point cloud, and to restore the compressed point cloud model. The proposed L3A measuring vector similarity was adopted. The CVS regarded each point as a three-dimensional vector connecting its coordinates and origin, and the reference vector was selected according to the reading order of the points (Long et al., 2017). Thus, the sampling area that could cover the whole point cloud was generated for partition compression (Xukang, Baoning & Jinwen, 2020). Qi et al. (2016) used the advantages of the tree structure of octree in spatial decomposition, combined with the bounding box simplification algorithm of the point cloud. So, the simplification of PCD was realized. Although this algorithm had certain speed advantages, it retained less detailed feature points.

In point cloud recognition, the target detection and recognition system could judge the position of multiple objects in the image and determine their categories according to the input image information (Chenhang et al., 2020; Kiyosumi et al., 2011; Yang, Luo & Urtasun, 2018; Chen et al., 2017). Before 2012, the primary method used for target detection was manually extracting relevant features based on human beings’ prior knowledge, such as HOG detection, DPM algorithm, etc. Although such algorithms could recognize target objects, the accuracy and operation time must be improved. In addition to the two-stage detection algorithm, some scholars studied the one-stage detection algorithm. In 2015, Joseph et al. (2016) proposed the YOLO algorithm, which could directly give the class probability and position information of objects through images instead of having to form a series of candidate boxes to classify them, similar to the two-stage algorithm, which significantly reduced the running time of the algorithm. This was the first one-stage algorithm. Afterwards, the YOLO algorithm was improved, resulting in YOLOv2 and YOLOv3 algorithms, further enhancing the algorithm’s running speed and detection accuracy. In 2017, Qi et al. (2017) proposed the PointNet network, which could directly use the original point cloud as input, avoiding the process of voxelization or projection of the original point cloud and significantly saving resources. However, PointNet has focused on global features and lacks local features.

The search for a method compressing PCD and using it to recognize a target object has also been targeted by experts in many fields of autonomous driving research. For example, Wiesmann et al. (2022) proposed a new neural network architecture for mapping data, which fully used the compression feature to reduce the storage space required for offline maps greatly and could transmit efficiently. They obtained a highly descriptive feature expression of the point cloud, which could be used for position recognition tasks.

After analyzing the limitations of the mentioned algorithms, this article proposes a PCD compression method based on feature point extraction. The method includes three steps: 1. to estimate the average vector and curvature value of each point according to its K neighbourhood, 2. to extract feature points for areas with significant changes in the angle between the curvature value and the average vector, and then use the nearest point of the centre point in the grid to replace other points in the flat area, and 3. to merge the two points and delete the duplicate points, to achieve the simplification of the point cloud.

This method could retain most of the feature points of the point cloud, and the detailed information is not lost, achieving the removal of redundant data and maximizing the quality of the PCD. Furthermore, the compressed PCD could improve the recognition accuracy of pedestrians.

Compression method based on boundary extraction

The research on point cloud compression is mainly divided into two directions. One is to reduce the number of point clouds by establishing a spatial structure by employing a random sampling scheme. The other is to compress the attributes of point clouds, including colours, normal vectors, reflection coefficients, etc. Thus, its purpose is to reduce the necessary storage capacity occupied by PCD and improve its storage and transmission efficiency utilization. The main content of this article is to achieve point cloud compression by reducing the number of points.

Point cloud compression method based on boundary extraction

Calculation of normal vector and curvature

The average vector and curvature value of PCD are the geometric attribute information. In feature point extraction, it is necessary to calculate the average vector and curvature value of the point cloud to retain more feature points.

After completing the K-neighborhood-neighborhood search of all points, the principal component analysis (PCA) can be used to calculate the normal vector of the point cloud (Xijiang, Guang & Xianghong, 2015). The normal vector of a tangent plane on the surface of the query point pi pi is approximately estimated. The space plane equation of the point fitting in its k k neighbourhood is defined by

(1) ax+by+cz−d=0

where a,b,c a,b,c denote the coefficients of the plane equation and also represent the normal vectors of the query point pi pi, x,y,z x,y,z represent the three-dimensional coordinate values of the midpoint of the k neighbourhood; d d denotes the distance from the origin to the plane. The covariance matrix, A A, is computed from the k-neighbourhood element of the query point p as follows:

(2) A=∑j=0k(pj−p¯)(pj−p¯)T

where k k denotes the number of adjacent points of point pi pi, p¯ p¯ represents the three-dimensional centroid of adjacent element coordinates. The covariance matrix A has three real eigenvalues: λ1 λ1, λ2 λ2, λ3,(λ1≤λ2≤λ3λ3), (λ1≤λ2≤λ3), the corresponding three eigenvectors are denoted by n1,n2,n3 n1,n2,n3, n1 n1 denotes the normal vector (nx,ny,nz) (nx,ny,nz). Then the curvature value of the query point pi pi is calculated by k k-neighbourhood surface fitting (Yu et al., 2010). The quadric surface function has universal applicability and is convenient for subsequent curvature calculation. Therefore, this article uses quadric surface fitting to calculate the curvature value. The initial fitting function of the quadric surface is generally defined by

(3) z=f(x,y)=ax2+bxy+cy2+ex+fy

where a, b, c, e, and f denote the coefficients of the quadratic surface fitting. Equation (3) shows a single-valued function, and a set of values corresponds to a unique z value. However, many non-single-value mapping problems exist in the collected data (Xin & Jingying, 2020). Therefore, it is necessary to establish a local coordinate system with the query point p as an origin (u,v,ω) (u,v,ω). The direction of the ω ω axis and the average vector at pi pi are the same. The u u and the v v axes are orthogonal in the tangent plane. Together with the ω ω axis, they form a rectangular coordinate system. After running both translations and rotations, the local coordinate system was established. Equation (3) is then changed to the parameter expression in local coordinates as follows:

(4) s(μ,ν)=(μ,ν,ω(μ,ν))

(5) ω(μ,ν)=aμ2+bμν+cν2+eμ+fν

Since the number of points in the k k neighbourhood is generally greater than 5, the coefficient of the surface equation could be resolved according to the least square method, and the curvature value of the point pi pi could be computed according to the relationship between the surface equation and the curvature defined by

(6) Hi=a+c

The extraction of boundary points

Extraction principle: for scattered PCD, if the point pi pi is a boundary point, the points in the neighborhood of point k k will be biased to one side in the 3D coordinate system. On the contrary, pi pi is called the internal point. This feature of boundary points could be used to identify boundary points according to the angle between the normal vector formed by the neighbourhood points and the query point pi pi. These steps are as follows:

The query point pi pi is used as an origin to establish the local coordinate system (u,v,ω) (u,v,ω). The direction of the coordinate system for the ω ω axis is the same as that of the normal vector at pi pi. The u u and the v v axes are orthogonal to each other in the tangent plane and form a rectangular coordinate system together with the ω ω axis. (1) With a pi pi as the reference point and the point pj pj in the neighbourhood of k k, the vector vij(j = 1,2,…,k) is established and projected onto the local coordinates established in the previous step to form the projection vector vj.

(2) Move in the counterclockwise direction, calculate the included angle βj(j+1)(whenj=k,j+1=1) βj(j+1)(whenj=k,j+1=1) of two adjacent projection vectors respectively, then sort the included angles βj(j+1) βj(j+1) in ascending order, and calculate the maximum difference Lmax of the two adjacent included angles.

(3) Determine whether it is a boundary point by comparing the value of the given angle thresholds Lstd and Lmax If Lstd>Lmax, pi pi denotes the boundary point, and vice versa, represents the internal point.

The extraction of sharp points

The steps are as follows:

(1) In the k k neighbourhood of pi pi, the weight of local curvature could be calculated according to the curvature value, and the expression is defined by

(7) {δi=1k∑j=0k(Hpi−H¯)2+|Hi=H¯|H¯=1k∑j=0kHpi

where k k denotes the number of point clouds in the PII neighbourhood; Hij Hij shows the curvature value of pj pj in the neighbourhood; IiI Hi represents the curvature value of the pi pi, and H¯ H¯ denotes the mean of the curvature of the pj in the neighbourhood.

The weight parameter of the local curvature is calculated based on the curvature value in the neighbourhood. It results from the joint contribution of multiple points and is not sensitive to a single noise point. Therefore, it has certain robustness.

When δi δi denotes a relatively small value, the geometric characteristics of the region are relatively gentle. On the other hand, when δi δi is relatively large, it means that the curvature value of the region changes greatly. The point pi pi has more geometric information in the region, and the greater the possibility of becoming a sharp point.

(2) The average of the angle between the normal vector in the query point pi pi and its k k neighbourhood could be calculated by Eq. (8)

(8) αi¯=1k∑j=0kαj

where αi αi denotes the angle between the normal vector ni of the point pipi and the normal vector nj of the point pj pj in the neighbourhood.

The included angle parameter considers the comprehensive influence of all points on the local surface change in the neighbourhood. Generally, the larger the value of αi¯ αi¯, the greater the change in the local area at which the point pi pi is located, and the higher the possibility that the point is sharp on the surface. On the other hand, the smaller the value of αi¯ αi¯, the more gentle the point cloud model in the region is, and the possibility that the point pi pi is a sharp point will also decrease.

The local curvature weight δiI calculated in Eq. (7) and the average of the included angle of the normal vector. αi¯ αi¯ in Eq. (8) are called the sharp point parameters. To avoid identification of the points composed of uneven distribution of the point cloud and the determination of the flat surfaces as sharp points, the average of the neighborhood distance is added to the definition of the sharp point parameters to improve the accuracy, so the sharp point parameters are defined by

(9) {fi=λ.δi+τ.αi¯di¯di¯=1k∑j=0kdj

where λ λ denotes the control coefficient of local curvature weight δi δi; τ represents the control coefficient of the average of the included angle for the normal vector αi¯ αi¯, di¯ di¯ denotes the average distance between pi pi and pj pj in its neighbourhood.

To avoid a sharp detection threshold, the control coefficient and τ control sharp point parameter are needed since F is difficult to determine. The recognition threshold of the feature point is defined by

(10) F=Hmaxt

where Hmax denotes the maximum curvature value, and t represents the distance values of all sharp points. If f > F, P is called a sharp point, otherwise a flat point.

The grid compression of point cloud

Grid compression is performed on the flat points in the PCD. The grid structure is established on the point cloud. Utilizing the centre point in the grid can improve the subsequent execution efficiency to a certain extent. However, the centre point has not necessarily been the point in the original data, which has a specific impact on the accuracy of the point cloud. So, similar to the nearest neighbour point based on the octree voxel centre point, this section uses the nearest neighbour point of the centre point in the grid to replace other points. To achieve the simplified representation of PCD and improve efficiency, the grid is directly built on it without using the structure of an octree, and the nearest neighbour of its centre point is taken. Finally, the extracted boundary of the point cloud and the two segmentations after grid simplification are combined to complete the operation of point cloud compression.

Point cloud pedestrian recognition algorithm based on image mapping

YOLO algorithm is an end-to-end two-dimensional target recognition model (Yujie, Xuanpeng & Weigong, 2020). YOLOv5 is a newer generation of the YOLO series of target recognition algorithms, which considers recognition accuracy and recognition speed concurrently. It is an efficient target recognition algorithm implemented in single-stage recognition. In the code of the YOLOv5 model, the four versions of detection networks are called YOLOv5x, YOLOv5l, YOLOv5m, and YOLOv5s, respectively, which could meet the use of different levels of devices. Besides, YOLOv5s is the network with the minor depth and width of the feature map in the series. YOLOv5 follows some methods of YOLOv4. For example, the Mosaic method is used for data enhancement. Different initial prior frame (anchor) aspect ratios are computed for different data sets. In the test phase, the last frame is added to the original image to improve the reasoning speed. Improved loss function called Generalized Intersection over Union (GIOU) Loss and distance—IoU (DIOU) are filtered by prediction box_ NMS, etc.

The network structure of the YOLOv5

YOLOv5 network structure comprises input, backbone, neck, and prediction components. The main functions of each component are described as follows: (1) Input: YOLOv5 uses the Mosaic data enhancement method in YOLOv4 to enrich data sets, reduce hardware requirements and reduce the use of GPU. The adaptive anchor frame calculation function is embedded in the whole training code. The switch can be adjusted automatically, and the adaptive image scaling can improve the speed of target detection in the reasoning stage.

(2) Backbone: The focus module is the unique structure of the YOLOv5. The key is to slice the feature map into blocks and then transfer the image features to the next block through multi-layer convolution pooling, cross-stage partial network (CSPNet) (Chien-Yao et al., 2020), and feature pyramid pooling (SPP) (He et al., 2015) structures.

(3) Neck: The feature pyramid networks (FPN) (Tsung-Yi et al., 2017) and path aggregation network (PAN) (Shu et al., 2018) structures of the YOLOv4 are used.

However, the CSP2 structure designed concerning CSPnet is adopted, which strengthens information dissemination and can accurately retain spatial information as the neck structure of the YOLOv5 differs.

(4) Prediction: The improved loss function called GIOU-Loss can accurately identify some objects with overlapping occlusion and resolve the defects of the conventional IOU.

The loss function of the YOLOv5 network

In the YOLOv5 network, CIoU CIoU loss in the IoU IoU series loss function is used to calculate the boundary box loss (Box) of the target box and prediction box. The standard and prediction frames’ confidence loss and classification loss are calculated by using the binary cross entropy loss (BCEL) function defined by (1) Bounding box loss:

(11) CIoU=IoU−ρ2c2−αν

(12) ν=4π2(arctan⁡ωgthgt−arctan⁡ωphp)2

(13) lossCIoU=1−CIoU

where the IoU IoU denotes the ratio of the overlapping area of the prediction box and the target box to the area of the parallel part; ρ denotes the distance between the centre point of the prediction box and the target box; ν is used to indicate the similarity of aspect ratio; α represents the influence coefficient of ν. (2) Confidence loss and classification loss:

(14) Loss=−∑i=1N(yilog⁡(p(xi)+(1−yi)log⁡(1−p(xi)))

where y denotes the real label and p(x) represents the model output.

Mapping of point cloud data to images

The acquisition platform of the Karlsruhe Institute of Technology and Toyota Technological Institute (KITTI) dataset is equipped with a camera, 64-line lidar, and optical lens (Zhiyong, Luo & Dai, 2017). Mapping the PCD to the rotation matrix of the image requires the calibration parameters of the lidar and camera, the internal parameters of the camera, and the rotation matrix. According to the conversion formula calibrated during the acquisition of the KITTI data set, the correspondence is computed between X(x,y,z,1) the point cloud coordinate and Y(u,v,1) image coordinate system, then, one-to-one correspondence is established between the PCD and the image pixel. Equation (15) expresses this relationship.

(15) Y=Prect(i)Rrect(0)TvelocamX

where Prect(i) Prect(i) denotes the internal parameter matrix representing the ith camera, Rrect(0) Rrect(0) represents the correction matrix from each camera to camera 0, Tvelocam Tvelocam denotes the rotation and translation matrices from the camera to the laser radar. Mapping image pixels to PCD requires each matrix’s row and column expansion. X represents the homogeneous coordinate form of PCD, and Tvelocam Tvelocam denotes the external parameter matrix of laser radar and the camera obtained through calibration, including the rotation matrix and translation matrix.

(16) α.(uv1)=(P00P10P20P30P01P11P21P31P02P12P22P32P03P13P23P33)∗(r00r10r200r01r11r210r02r12r220r03r13r231)∗(t00t10t200t01t11t210t02t12t220t03t13t231)∗(xyz1)

According to Eq. (16), points of PCD can be mapped to image coordinates through the conversion matrix between lidar and camera. The image with YOLO network recognition is taken as the input. The point cloud can be mapped to the image with the target bounding box. Pixel points in the bounding box are justified through coordinate transformation and mapped to the points in the corresponding PCD. Afterwards, the point cloud coordinates mapped to the image in the bounding box are judged. The point cloud coordinates corresponding to the pixels in the bounding box are counted, which is used to calculate the three-dimensional bounding box of a pedestrian in the PCD. Thus, the position of a target pedestrian in the point cloud is detected with a higher recognition rate.

K-means clustering of point cloud pedestrian

The K-means algorithm is a relatively simple unsupervised clustering approach (Jinghui, Wei & Ru, 2020). The algorithm’s input is a sample set divided into K subsets. K initial cluster centres are randomly selected, and then the distance between each sample and the cluster centre is calculated. The cluster results are output when the partition of the subsets satisfies certain requirements. In this article, the PCD is used as the input, Euclidean distance is utilized as the clustering condition, and the pedestrian recognition results of two-dimensional images are employed as a priori knowledge to cluster the pedestrians satisfying the requirements of clustering in the point cloud scene. The K-means clustering is implemented in the point cloud.

For a given sample set divided into K clusters according to the size of the sample point distance, the set of clusters is denoted by (C1,C2,⋯⋯Ck), and the iterative method is used to minimize the squared error defined by Eq. (17)

(17) E=∑i=1k∑X=Ci‖x−ui‖22

(18) ui=1|Ci|∑x∈Cix

where ui denotes the mean vector of Ci and is denoted by Eq. (18). The distance from the point to all centre points belonging to the clusters of the nearest center point is calculated for each point. After running iterative calculations, the centre point of the cluster is recalculated, and the centre point nearest to each point is found until there is no change in the previous two iteration results.

In K-means clustering, determining the initial mean vector for K clusters and selecting the location will affect the pedestrian clustering results and running time (Wang, Gao & Feng, 2020). The conventional K-means clustering selects the mean vector completely at random, which will slow the convergence speed of the algorithm. Therefore, choosing the appropriate mean vector and the number of clusters, K, is necessary. The commonly used methods to determine K include the elbow and contour coefficient methods (Pei & Rui, 2015). The elbow method uses the sum of the square errors (SSE) to select the appropriate K defined by Eq. (19). As K increases, the sample division will become more and more refined, and the aggregation degree of each cluster will gradually increase. The core index of the contour coefficient method is called the Silhouette coefficient (SC), defined by Eq. (20). The contour coefficient determines the nearest cluster.

(19) SSE=∑i=1k∑p∈Ci|p−mi|2

(20) S=b−amax(a,b)

When K is determined by elbow and contour coefficient methods, there will be a clustering error of the pedestrian target in the point cloud. To overcome the problem of pedestrian misrecognition caused by the point cloud clustering, the K-means clustering is optimized by using prior knowledge of image recognition. First, the input PCD is obtained through image mapping. The two-dimensional recognition of prior coordinates for the data set is carried out. The number of pedestrians in the image recognition is taken as K through cross-validation. Then, to reduce the computational complexity, the mean vectors of K clusters are determined by mapping the midpoint of the two-dimensional boundary box (Xin & Jingying, 2020). Because it is uncertain whether the bounding box’s centre point pixel corresponds to the PCD, the PCD corresponding to the pixel in the bounding box is randomly selected as the mean clustering vector.

K and the mean vectors of K clusters are determined. The pedestrian clustering algorithm based on K-means clustering is presented as follows:

(1) After mapping the image to the point cloud, determine the input PCD D={x1,x2,…xm} D={x1,x2,…xm}, and use the K-means clustering to classify pedestrians on the point cloud.

(2) Select the appropriate K. The prior knowledge of image recognition is used to select the number of final bounding boxes in the pedestrian recognition of the image as K.

(3) The centre of clustering is set by selecting K mean vectors {u1,u2,…uk} {u1,u2,…uk} from the PCD. The points in the bounding image box are mapped as point clouds as the mean vectors of clusters are selected randomly, and one mapping pixel point in each bounding box is selected.

(4) The PCD is iteratively divided and initialized to C={C1,C2,…Ck} C={C1,C2,…Ck}, and the distance between the sample point xi(i=1,2,3,…m} xi(i=1,2,3,…m} and the initial mean vector uj{j=1,2,3,…,k} uj{j=1,2,3,…,k} of the input point cloud is calculated by Eq. (21).

(21) Dij=‖xi−uj‖22Dij=‖xi−uj‖22

Put xi xi Then, update the cluster into the category where the distance becomes the smallest with the mean vector.

(22) Cλi=Cλi∪{xi} Cλi=Cλi∪{xi}

(5) Recalculate the mean vector for all points in Cj, as shown in Eq. (23).

(23) uj=1|Cj|∑x∈Cjx uj=1|Cj|∑x∈Cjx

(6) Judge whether all mean vectors have changed. If there is no change, output the clustered point cloud C={C1,C2,…Ck}.

Experiment and the analysis of the results

Basic idea of the compression method

The PCD of three groups of pedestrian scenes on the road in the KITTI data set (format: pcd file, file name: 003922. pcd, 003955. pcd, 003971. pcd) were used as the experimental objects. The three data groups were named pedestrian detection scene on road 1, pedestrian detection scene on road 2, and pedestrian detection scene on road 3.

The PCD used in the experiment contains rich feature information, including both flat points with slow curvature change and feature points with special visual effects. In the experimentation, the normal vector and curvature of the point cloud are calculated, and then the boundary lines and sharp points are extracted. The process of grid reduction on the PCD of other regions is carried out. Finally, the boundary, sharp, and processed points of the grid reduction are fused. Moreover, the duplicate data is deleted. The entire experiment is completed by saving the compression results.

The algorithm consists of two modules: the compression and recognition modules. The basic idea of the compression module is as follows:

Algorithm (Compression)

Read the point cloud, and establish the topological structure for the point cloud.

Calculate the normal vector and curvature of the point cloud.

Extract the feature points and sharp points of the point cloud.

Use the nearest neighbour point of the centre point in the grid to replace other points in the flat area and simplify the grid processing. The PCD in the flat area is obtained when calculating sharp points.

Fuse the PCD and delete the same points.

Save the compressed PCD.

The basic idea of the recognition module includes three parts:

Algorithm (Recognition)

Use YOLO to recognize pedestrians’ images.

Construct a mapping between PCD and image,

Employ K-means clustering to detect targets.

YOLOv5 is used as the basic network for image recognition. The pixel points in the bounding box of the target image corresponding to the PCD to narrow the range of the point cloud target. The output of the pedestrian target bounding box filters the pedestrian in the point cloud (Zanfeng, 2012). After conducting image-to-point cloud mapping, there would inevitably be interference data in the PCD, so it is necessary to process the mapped PCD. The K-means clusters the mapped point cloud to identify pedestrians in the PCD.

The experimentation for the attribute compression of the point cloud is carried out. Using the platform tmc13, the data set’s three groups of test sequences are tested separately. After configuring the environment, the MPEG-based codec software is compiled and debugged. The environment configuration of the experimental platform is shown in Table 1.

Table 1 Experimental environment configuration.

Name	Type	
Processor	Inter(R) Core(TM) i5-9300HF CPU @2.40 GHz 2.40 GHz	
Graphics card	NVIDIA GeForce GTX1650	
Reference software	TMC13-v3.0	
Test sequence	Pavement pedestrian detection scene 1, scene 2, scene 3	
Codec configuration	PositionQuantizationScale:1, bitdepth:8	

The processing of the experimental compression

The results of the experimental compression are as follows:

Table 2 shows the change in the number of point clouds in each process and the final compression rate and the number of boundary points in the point cloud model, which is the smallest. Because of the many surface features of the point cloud model tested in this article, the number of sharp points extracted also grows larger. Extracting these sharp points can ensure that the geometric features of the point cloud are not easily lost while achieving a higher compression rate. To verify the reliability of the compression algorithm in this article, the random sampling method is used for comparative experimental analysis by using the same compression rate. The compression rates used in the experiments for the three groups of PCD are 37.95%, 44.32%, and 36.40%, respectively.

Table 2 Number of point cloud variables in each process of three groups of pavement pedestrian point cloud data compression.

Point cloud objects	Original point cloud per object	Boundary point cloud per object	Sharp point per object	Compress grid point clouds per object	Compression result per object	Compression ratio %	
Detection scene 1	115,119	1,496	50,116	28,851	71,431	37.95	
Detection scene 2	122,421	2,081	36,916	36,179	68,164	44.32	
Detection scene 3	123,843	2,972	58,132	23,576	78,764	36.40	

Comparative analysis of the evaluation of the experimental surface area

The total area of the triangulation constructed by the point cloud before and after compression is calculated. The change in the entire area is computed, and the change of the point cloud model features before and after compression is judged. The larger the change rate, the more feature points are deleted during compression. On the contrary, most of the deleted data are redundant, and the model’s features have little change.

The area of the PCD of the three groups based on the experimental objects is computed. The area of pedestrian detection scene I before compression is 27,635.62 cm2, the area of pedestrian detection scene 2 before compression is 51,008.45 cm2, and the area of pedestrian detection scene III before compression is 62,835.28 cm2. Table 3 lists the changes in the area.

Table 3 The area change of three groups of subjects after compression by two methods.

Point cloud objects	Compression method	Compressed area/cm2	Area of change/cm2	Area change rate %	
Detection
scene 1	The method proposed in this article	27,347.3	288.32	1.04	
Random sampling method	27,116.96	518.66	1.88	
Detection
scene 2	The method proposed in this article	50,624.29	384.16	0.75	
Random sampling method	50,508.19	500.26	0.98	
Detection
scene 3	The method proposed in this article	61,738.5	1,096.78	1.7	
Random sampling method	61,518.67	1,316.61	2.1	

Table 4 shows that the compression rate is the same, and the point cloud surface area change rate after compression is the smallest among the three compression methods. The value itself is relatively small, achieving the goal of a higher compression rate and smaller accuracy loss. During the compression process, not many feature points are not deleted, and the geometric features of the point cloud model are well preserved.

Table 4 Comparison of pedestrian recognition accuracy of different algorithms in KITTI dataset.

Algorithm	Data	Simple	Medium	Difficult	
Point pillars	Point Cloud	78.35%	72.26%	69.12%	
YOLO-v5	Image	96.34%	90.78%	86.56%	
Method in
this article	Point Cloud +Image	95.28%	89.46%	85.24%	

Evaluation index of point cloud compression experiment

For the compression of point cloud attributes, the compression mode is called losss-geom-lossy-attrs. This article will quantitatively analyze the experimental results by using the mean square error (MSE) and peak signal-to-noise ratio (PSNR) indicators. The content of the point cloud compression method based on boundary extraction will be directly compared with the experimental results of the corresponding literature. However, the method evaluating the quality of the point cloud will not be used. PSNR is usually used in voice and image processing. PCD, as a new media form, is also applicable. MSE and PSNR are used to evaluate the processing results of the point cloud.

(24) PSNRA,B=10log10(psdA,BMSE)

(25) PSNR=max(PSNRA,B,PSNRB,A)

ps denotes the signal peak. The signal peak is expressed by the length of the diagonal, and the point coordinates are normalized within the range [0,1] in Eq. (24).

(26) LD=‖(xmax,ymax,zmax)−(xmin,ymin,zmin)‖2

The point cloud is voxelated, and the peak value of each coordinate is expressed in depth precision b bits:

(27) pc=2b−1

The signal peak in Eq. (24) is defined by

(28) ps=3pc

Equation (29) represents the distance from one point to another point. Specifically, it is the square of the coordinate distance of point i in point cloud A (original point cloud) and the coordinate distance of point j in point cloud B (distorted point cloud) that is closest to point i (usually understood as the corresponding point- the modulus of vector (i, j)).

Equation (30) represents the distance from the point to the plane, that is, the length of the projection of the error vector (i, j) in the direction for the lower surface of a normal vector of the j point in the B point cloud, which is also the same value, making the PSNR of the point cloud have a different direction (A to B or B to A).

Finally, the PSNR value of the point cloud can be obtained by substituting the calculation results into Eqs. (24) and (25).

The PCD before and after filtering is represented by P P and P∗ P∗ respectively, the three-dimensional coordinate values of P P and P∗ P∗ are represented by (xP,yP,zP) (xP,yP,zP), (xP∗,yP∗,zP∗) (xP∗,yP∗,zP∗), and the total number of surface vertices is denoted by |{P}| |{P}|.

(29) dA,BP02P0=‖e→(i,j)‖22

(30) dA,BP02Pl=‖e^(i,j)‖22=(e→(i,j).nj→)2

The comparison of mean square error (MSE) and peak signal-to-noise ratio (PSNR) of two methods for the 1st pedestrian detection scenarios on the road is shown in Figs. 1A and 1B.

Figure 1 MSE and PSNR of the 1st pedestrian detection scenario on the road.

The comparison of mean square error (MSE) and peak signal-to-noise ratio (PSNR) of two methods for the 2nd pedestrian detection scenarios on the road is shown in Figs. 2A and 2B.

Figure 2 MSE and PSNR of the 2nd pedestrian detection scenario on the road.

The comparison of mean square error (MSE) and peak signal-to-noise ratio (PSNR) of two methods for the 3rd pedestrian detection scenario on the road is shown in Fig. 3.

Figure 3 MSE and PSNR of the 3rd pedestrian detection scenario on the road.

The smaller the MSE, the better the point cloud compression effect, and the larger the PSNR, indicating that the smaller the distortion of the point cloud model, the higher the matching degree with the original point cloud model, and the higher the overall compression quality of the point cloud. The results indicate that when the compression rate exceeds 30%, the proposed method performs better than the other method. When the compression rate is 90%, the advantage reaches the maximum, about 5.02% higher than the random sampling algorithm.

The MSE of the proposed method is smaller than that of the other method, regardless of the compression ratio. The proposed method is also superior to the other method when PSNR is used. When the compression ratio is 90%, the advantage grows larger, indicating that the proposed algorithm has certain advantages over other algorithms.

Comparison of the accuracy of image recognition algorithm after point cloud compression

To verify the effectiveness of the proposed algorithm in point cloud pedestrian recognition, the point cloud pedestrian recognition algorithm using voxel division, such as point pillars, and the image-based pedestrian recognition algorithm YOLO are compared. The outcomes are shown in Figs. 4–6, respectively.

Figure 4 Recognition results of the 1st pedestrian detection scene on the road.

Figure 5 Recognition results of the 2nd pedestrian detection scene on the road.

Figure 6 Recognition results of the 3rd pedestrian detection scene on the road.

First, the algorithms that take PCD and images as input are compared. The point-pillars method uses cylindrical elements to divide PCD and uses 2D convolution instead of a 3D convolution network. The speed of point cloud target recognition is enhanced, and the problem of long-distance pedestrian target recognition is resolved to a certain extent. Based on the mapping image to point cloud target recognition algorithm, the pedestrian target is recognized in the image data, which can recognize the pedestrian target at a relatively long distance. Then, the target object point cloud is quickly filtered through the mapping matrix between the PCD and the image. Through the optimization of the K-means clustering and the proposed algorithm, the target in the point cloud scene could be quickly determined by selecting the target and calculating the bounding box through the image recognition algorithm. The proposed algorithm inputs images and PCD and compresses them before doing the recognition stage, which can realize the recognition of pedestrians in the point cloud scene. However, since both PCD and compression processing are introduced, its accuracy may deteriorate in mapping point cloud images, and the recognition accuracy would become slightly lower than that of the image recognition algorithm.

However, introducing point cloud scenes breaks through the two-dimensional limit and has a wider application range than the simple image recognition algorithm. The experimental results show that the algorithm proposed in the manuscript is only inferior to the image-based pedestrian recognition method and is significantly superior to the above pedestrian recognition method based on PCD. In addition, this article uses the calibration matrix of PCD and image to filter the target point cloud, which largely inherits the accuracy of the image recognition algorithm and has advantages in pedestrian recognition of point cloud scenes.

Conclusion

This article proposes a point cloud data compression method based on feature point extraction and reduced voxel grid. The proposed method extracts the boundary and sharp points of the point cloud in the feature area and reduces the grid in the flat area. Finally, PCD compression is realized.

In the experiments, three groups of PCD, including pavement pedestrian information in the KITTI data set, are used. The results show that the proposed method could achieve data compression better and ensure that many feature points are retained in the compressed PCD. Thus, the compressed PCD achieves pedestrian recognition through an image-based mapping recognition algorithm.

When compared with the method of pedestrian recognition based on an image mapping recognition algorithm after running data compression with random sampling, the results show the reliability of the proposed method. In addition, in extracting sharp points in the compression algorithm proposed in this article, the control coefficient of the local curvature weight and the average control coefficient of the included angle must be set. Because the judgment threshold of sharp points has the maximum curvature value, the two coefficients could be selected in an extensive range. In addition, to simplify the operation, this article selects grid simplification in the flat area of the point cloud.

Future work plans to improve further the simplicity of the proposed method to automatically determine the control coefficient of the sharp point extraction value, the local curvature weight value, and the average control coefficient of the included angle. Moreover, better compression results could be attained by optimizing the compression of the flat area. In addition, the experimental objects used in this experiment are insufficient to cover all the real road conditions. Therefore, more complex samples would be used to optimize the algorithm.

Supplemental Information

Supplemental Information 1 Code.

Click here for additional data file.

Additional Information and Declarations

Competing Interests

Author Contributions

Data Availability

The authors declare that they have no competing interests.

Yanjun Zhang conceived and designed the experiments, performed the experiments, analyzed the data, performed the computation work, prepared figures and/or tables, authored or reviewed drafts of the article, and approved the final draft.

The following information was supplied regarding data availability:

The code is available in the Supplemental File and the data is available at 3DCOMET: http://www.rovit.ua.es/dataset/3dcomet/downloads.html.

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
