# Peer review of "A novel fast pedestrian recognition algorithm based on point cloud compression and boundary extraction"

_PeerJ Computer Science, doi:10.7717/peerj-cs.1426_

## Round 0.1 · original submission · Major Revisions

The reviewers are suggesting major revisions with regard to your article, therefore carefully revise your paper and resubmit. Please also improve the language of your manuscript professionally

Reviewer 1 ·

Basic reporting

We do not underline every single issue regarding the language and presentation. However, a complete revision is required. So, a complete language check is needed to resolve the issues of sentence structure, meaning, word order, utilization of formal words, vocabulary, misspellings, punctuation, and presentation of the conducted research.

Experimental design

All sections should be rewritten, reorganized, and shortened. The conclusion section should be rewritten, and the logical order should reconstruct the following: 1. the research motivation, 2the method proposed, 3 the contribution, 4. The advantages and disadvantages of the proposed method based on the literature review, 5. Key findings, and finally 6. Future research.

Validity of the findings

The technical competency of the article needs some revisions and reorganizations as follows:

1. The proposed algorithm should be presented
2. At what ratio does the compression algorithm deteriorate? Do they run any studies?
3. Results should be better presented and discussed
4. Do the authors run any experiments for the number clusters (K) and the mean vector in K-means clustering?
5. There are algorithms more improved than K-means. Why did the authors choose it?
6. We believe that it is helpful to add a table that presents MSE and PSSR results.
7. In Table 5-4, the score of the proposed method is slightly less than the YOLOv5 algorithm in 3 types of data sets. Are these results a disadvantage for the proposed method? Please discuss it.

Additional comments

The paper needs to be revised in major aspects

·

Basic reporting

Even though this paper proposes a point cloud data compression method based on feature point extraction and contributes to the literature, it has several issues regarding the presentation, language, and clarity of the proposed method.

The language and presentational issues that need to be fixed are as follows:

1. Abstract should be rewritten by underlying those as follows: 1. What is the motivation of the research? 2. What are the data and method/model implemented? 3. What is the contribution of this research? 4. What are the key results of this research?
2. Introduction section should be shortened and rewritten. More references should be added. Two more paragraphs should be added at the end of the introduction section. 1. the research motivation and contribution. 2. how the rest of the article is structured.
3. Section 3 should be rewritten and reorganized. Eqs(.) should be cited in the text. The term “formula” should be disregarded.
This sentence is extracted from the text: “Take the local curvature weight δ_i calculated in (1) and (2)”
What do (1) and (2) refer to?
4. This sentence is extracted from the text: “often at the G level, even reaching the T level,”
What are the T and G levels?
5. The titles and subtitles of sections should be checked and corrected. All titles and subtitles should be bold.
6. These sentences are extracted from the text:
“This method lacks attention to feature points, resulting in that the data points retained after….”
“another part of scholars have studied the one-stage detection algorithm.”

Therefore, proofreading is a must.

7. The titles and subtitles of all tables and figures should be checked.
8. References in the text and the References section should be checked.
9. Section 3 should be rewritten and reorganized. This sentence is extracted from Section 3: “Create the covariance matrix A”. Generally, a covariance matrix is not created but computed.
10. References in the References section should be checked and corrected.
11. Sections and subsections should be bold. Their fonts should be larger.



Technical soundness of the article:

1. In subsection 3.2.1 Boundary extraction point, there are steps explained in order. However, the first step starts with (2). Please check and correct it.
2. Section 3 should be rewritten and reorganized.
3. In subsection 3.1.3, this sentence is extracted from the text: “Take the local curvature weight δ_i calculated in (1) and (2)”. What are (1) and (2)? Please clarify them.
4. Instead of using “formula (4-11).”, Eq.(.) should be used
5. This sentence is extracted from the paper: “The algorithm consists of two modules, namely compression module and recognition module.” Please provide both Algorithms
6. Please provide information on how random sampling is applied
7. The proposed algorithms should be presented.

Experimental design

Even though this paper proposes a point cloud data compression method based on feature point extraction and contributes to the literature, it has several issues regarding the presentation, language, and clarity of the proposed method.

The language and presentational issues that need to be fixed are as follows:

1. Abstract should be rewritten by underlying those as follows: 1. What is the motivation of the research? 2. What are the data and method/model implemented? 3. What is the contribution of this research? 4. What are the key results of this research?
2. Introduction section should be shortened and rewritten. More references should be added. Two more paragraphs should be added at the end of the introduction section. 1. the research motivation and contribution. 2. how the rest of the article is structured.
3. Section 3 should be rewritten and reorganized. Eqs(.) should be cited in the text. The term “formula” should be disregarded.
This sentence is extracted from the text: “Take the local curvature weight δ_i calculated in (1) and (2)”
What do (1) and (2) refer to?
4. This sentence is extracted from the text: “often at the G level, even reaching the T level,”
What are the T and G levels?
5. The titles and subtitles of sections should be checked and corrected. All titles and subtitles should be bold.
6. These sentences are extracted from the text:
“This method lacks attention to feature points, resulting in that the data points retained after….”
“another part of scholars have studied the one-stage detection algorithm.”

Therefore, proofreading is a must.

7. The titles and subtitles of all tables and figures should be checked.
8. References in the text and the References section should be checked.
9. Section 3 should be rewritten and reorganized. This sentence is extracted from Section 3: “Create the covariance matrix A”. Generally, a covariance matrix is not created but computed.
10. References in the References section should be checked and corrected.
11. Sections and subsections should be bold. Their fonts should be larger.



Technical soundness of the article:

1. In subsection 3.2.1 Boundary extraction point, there are steps explained in order. However, the first step starts with (2). Please check and correct it.
2. Section 3 should be rewritten and reorganized.
3. In subsection 3.1.3, this sentence is extracted from the text: “Take the local curvature weight δ_i calculated in (1) and (2)”. What are (1) and (2)? Please clarify them.
4. Instead of using “formula (4-11).”, Eq.(.) should be used
5. This sentence is extracted from the paper: “The algorithm consists of two modules, namely compression module and recognition module.” Please provide both Algorithms
6. Please provide information on how random sampling is applied
7. The proposed algorithms should be presented.

Validity of the findings

Even though this paper proposes a point cloud data compression method based on feature point extraction and contributes to the literature, it has several issues regarding the presentation, language, and clarity of the proposed method.

The language and presentational issues that need to be fixed are as follows:

1. Abstract should be rewritten by underlying those as follows: 1. What is the motivation of the research? 2. What are the data and method/model implemented? 3. What is the contribution of this research? 4. What are the key results of this research?
2. Introduction section should be shortened and rewritten. More references should be added. Two more paragraphs should be added at the end of the introduction section. 1. the research motivation and contribution. 2. how the rest of the article is structured.
3. Section 3 should be rewritten and reorganized. Eqs(.) should be cited in the text. The term “formula” should be disregarded.
This sentence is extracted from the text: “Take the local curvature weight δ_i calculated in (1) and (2)”
What do (1) and (2) refer to?
4. This sentence is extracted from the text: “often at the G level, even reaching the T level,”
What are the T and G levels?
5. The titles and subtitles of sections should be checked and corrected. All titles and subtitles should be bold.
6. These sentences are extracted from the text:
“This method lacks attention to feature points, resulting in that the data points retained after….”
“another part of scholars have studied the one-stage detection algorithm.”

Therefore, proofreading is a must.

7. The titles and subtitles of all tables and figures should be checked.
8. References in the text and the References section should be checked.
9. Section 3 should be rewritten and reorganized. This sentence is extracted from Section 3: “Create the covariance matrix A”. Generally, a covariance matrix is not created but computed.
10. References in the References section should be checked and corrected.
11. Sections and subsections should be bold. Their fonts should be larger.



Technical soundness of the article:

1. In subsection 3.2.1 Boundary extraction point, there are steps explained in order. However, the first step starts with (2). Please check and correct it.
2. Section 3 should be rewritten and reorganized.
3. In subsection 3.1.3, this sentence is extracted from the text: “Take the local curvature weight δ_i calculated in (1) and (2)”. What are (1) and (2)? Please clarify them.
4. Instead of using “formula (4-11).”, Eq.(.) should be used
5. This sentence is extracted from the paper: “The algorithm consists of two modules, namely compression module and recognition module.” Please provide both Algorithms
6. Please provide information on how random sampling is applied
7. The proposed algorithms should be presented.

Additional comments

Even though this paper proposes a point cloud data compression method based on feature point extraction and contributes to the literature, it has several issues regarding the presentation, language, and clarity of the proposed method.

The language and presentational issues that need to be fixed are as follows:

1. Abstract should be rewritten by underlying those as follows: 1. What is the motivation of the research? 2. What are the data and method/model implemented? 3. What is the contribution of this research? 4. What are the key results of this research?
2. Introduction section should be shortened and rewritten. More references should be added. Two more paragraphs should be added at the end of the introduction section. 1. the research motivation and contribution. 2. how the rest of the article is structured.
3. Section 3 should be rewritten and reorganized. Eqs(.) should be cited in the text. The term “formula” should be disregarded.
This sentence is extracted from the text: “Take the local curvature weight δ_i calculated in (1) and (2)”
What do (1) and (2) refer to?
4. This sentence is extracted from the text: “often at the G level, even reaching the T level,”
What are the T and G levels?
5. The titles and subtitles of sections should be checked and corrected. All titles and subtitles should be bold.
6. These sentences are extracted from the text:
“This method lacks attention to feature points, resulting in that the data points retained after….”
“another part of scholars have studied the one-stage detection algorithm.”

Therefore, proofreading is a must.

7. The titles and subtitles of all tables and figures should be checked.
8. References in the text and the References section should be checked.
9. Section 3 should be rewritten and reorganized. This sentence is extracted from Section 3: “Create the covariance matrix A”. Generally, a covariance matrix is not created but computed.
10. References in the References section should be checked and corrected.
11. Sections and subsections should be bold. Their fonts should be larger.



Technical soundness of the article:

1. In subsection 3.2.1 Boundary extraction point, there are steps explained in order. However, the first step starts with (2). Please check and correct it.
2. Section 3 should be rewritten and reorganized.
3. In subsection 3.1.3, this sentence is extracted from the text: “Take the local curvature weight δ_i calculated in (1) and (2)”. What are (1) and (2)? Please clarify them.
4. Instead of using “formula (4-11).”, Eq.(.) should be used
5. This sentence is extracted from the paper: “The algorithm consists of two modules, namely compression module and recognition module.” Please provide both Algorithms
6. Please provide information on how random sampling is applied
7. The proposed algorithms should be presented.

---

## Round 0.2 · accepted · Accept

Congratulations, your paper is acceptable after the improvement and subsequent recommendations by the experts

Reviewer 1 ·

Basic reporting

The article has been updated according to my previous comments. Therefore, I have no more comments and recommend it for publishing

Experimental design

The experimental design is updated and it seems to have sufficient technical innovation to be accepted

Validity of the findings

The findings are good enough

Additional comments

Overall im satisfied with the current version of the paper.

·

Basic reporting

no comment

Experimental design

no comment

Validity of the findings

no comment

Additional comments

Author has done the requested change. I think that the paper can be published in the present form